# INVARIANT-EQUIVARIANT REPRESENTATION LEARNING FOR MULTI-CLASS DATA

## ABSTRACT

Representations learnt through deep neural networks tend to be highly informative, but opaque in terms of what information they learn to encode. We introduce an approach to probabilistic modelling that learns to represent data with two separate deep representations: an invariant representation that encodes the information of the class from which the data belongs, and an equivariant representation that encodes the symmetry transformation defining the particular data point within the class manifold (equivariant in the sense that the representation varies naturally with symmetry transformations). This approach to representation learning is conceptually transparent, easy to implement, and in-principle generally applicable to any data comprised of discrete classes of continuous distributions (e.g. objects in images, topics in language, individuals in behavioural data). We demonstrate qualitatively compelling representation learning and competitive quantitative performance, in both supervised and semi-supervised settings, versus comparable modelling approaches in the literature with little fine tuning.

## 1 INTRODUCTION

Representation learning (Bengio et al., 2013) is part of the foundation of deep learning; powerful deep neural network models appear to derive their performance from sequentially representing data in more-and-more refined structures, tailored to the training task.

However, representation learning has a broader impact than just model performance. Transferable representations are leveraged efficiently for new tasks (Mikolov et al., 2013), representations are used for human interpretation of machine learning models (Mahendran & Vedaldi, 2015), and meaningfully structured (disentangled) representations can be used for model control (e.g. semi-supervised learning as in Kingma et al. (2014), topic modelling as in Blei et al. (2003)).

Consequently, it is often preferable to have interpretable data representations within a model, in the sense that the information contained in the representation is easily understood and the representation can be used to control the output of the model (e.g. to generate data of a given class or with a particular characteristic). Unfortunately, there is often a tension between optimal model performance and cleanly disentangled or controllable representations.

To overcome this, some practitioners have proposed modifying their model's objective functions by inserting parameters in front of particular terms (Bowman et al., 2016; Higgins et al., 2017), while others have sought to modify the associated generative models (Mansbridge et al., 2018). Further still, attempts have been made to build the symmetries of the data directly into the neural network architecture in order to force the learning of latent variables that transform meaningfully under those symmetries (Sabour et al., 2017). The diversity and marginal success of these approaches point to the importance and difficulty of learning meaningful representations in deep generative modelling.

In this work we present an approach to probabilistic modelling of data comprised of a finite number of distinct classes, each described by a smooth manifold of instantiations of that class. For convenience, we call our approach EQUIVAE for Equivariant Variational Autoencoder. EQUIVAE is a probabilistic model with 2 latent variables: an invariant latent that represents the global class information, and an equivariant latent that smoothly interpolates between all of the members of that class. The EQUIVAE approach is general in that the symmetry group of the manifold must not be specified (as in for example Cohen & Welling (2014); Falorsi et al. (2018)), and it can be used for

any number of classes and any dimensionality of both underlying representations. The price that must be paid for this level of model control and flexibility is that some labelled data is needed in order to provide the concept of class invariance versus equivariance to the model.

The endeavor to model the content and the style of data separately is certainly not new to this work (Tenenbaum & Freeman, 2000). Reed et al. (2014) and Radford et al. (2016) go further, disentangling the continuous sources of variation in their representations using a clamping technique that exposes specific latent components to a single source of variation in the data during training. In the same vein, other approaches have used penalty terms in the objective function that encourage the learning of disentangled representations (Cheung et al., 2014; Chen et al., 2016).

EQUIVAE does not require any modification to the training algorithm, nor additional penalty terms in the objective function in order to bifurcate the information stored in the two latent variables. This is due to the way in which multiple data points are used to reconstruct a single data point from the same-class manifold, which we consider the primary novel aspect of our approach. In particular, our invariant representation takes as input multiple data points that come from the same class, but are different from the data point to be reconstructed. This invariant representation thus directly learns to encode the information common to the overall class, but not the individual data point, simply due to the information flowing through it.

Of further note, we deliberately use a deterministic latent for the invariant representation, and a stochastic latent for the smooth equivariant representation (an idea also employed by Zhu et al. (2014)). This choice is why we do not need to explicitly force the equivariant latent to not contain any class-level information: it is available and easier to access from the deterministic latent.

EQUIVAE is also comparable to Siddharth et al. (2017), where the authors leverage labelled data explicitly in their generative model in order to force the VAE latent to learn the non-class information (Makhzani et al. (2016) do similarly using adversarial training). The primary difference between those works and ours is that EQUIVAE provides a non-trivial representation of the global information instead of simply using the integer-valued label. Furthermore, this invariant representation can be deterministically evaluated directly on unlabelled data. Practitioners can reuse this embedding on unlabelled data in downstream tasks, along with the equivariant encoder if needed. The invariant representation provides more information than a simple prediction of the class-label distribution.

The encoding procedure for the invariant representation in EQUIVAE is partially inspired by Eslami et al. (2018), who use images from various, known coordinates in a scene in order to reconstruct a new image of that scene at new, known coordinates. In contrast, we do not have access to the exact coordinates of the class instance, which in our case corresponds to the unknown, non-trivial manifold structure of the class; we must infer these manifold coordinates in an unsupervised way. Garnelo et al. (2018a;b) similarly explore the simultaneous usage of multiple data points in generative modelling in order to better capture modelling uncertainty.

## 2 EQUIVARIANT VARIATIONAL AUTOENCODERS

We consider a generative model for data comprised of a finite set of distinct classes, each of which occupies a smooth manifold of instantiations. For example, images of distinct objects where each object might be in any pose, or sentences describing distinct sets of topics. Such data should be described by a generative model with two latent variables, the first describing which of the objects the data belongs to, and the second describing the particular instantiation of the object (e.g. its pose).

In this way the object-identity latent variable $r$ would be *invariant* under the transformations that cover the set of possible instantiations of the object, and the instantiation-specifc latent variable $v$ should be *equivariant* under such transformations. Note that the class label $y$ is itself an invariant representation of the class, however, we seek a higher-dimensional latent vector $r$ that has the capacity to represent rich information relevant to the class of the data point, rather than just its label.

Denoting an individual data point as $x_n$ with associated class label $y_n$, and the full set of class-$y$ labeled data $\{x_n | \text{label}(x_n) = y\}$ as $\mathcal{D}_{\text{lab}}^y$, we write such a generative model as:

$$p\big(\{x_n, y_n\}_{n=1}^N\big) = \prod_{n=1}^{N} \int dv_n \, dr_n \, p_\theta(x_n | r_n, v_n) \, \delta\big(r_n - r(\mathcal{D}_{\text{lab}}^{y_n} \setminus \{x_n\})\big) \, p(v_n) \, p(y_n) \quad (1)$$

where $\theta$ are the parameters of the generative model. We make explicit with a $\delta$ function the conditional dependency of $p_\theta$ on the deterministically calculable representation $r_n$ of the global properties of class $y_n$. The distribution $p(y_n)$ is a categorical distribution with weights given by the relative frequency of each class and the prior distribution $p(v_n)$ is taken to be a unit normal describing the set of smooth transformations that cover the class-$y_n$ manifold.

To guarantee that $r_n$ will learn an invariant representation of information common to the class-$y_n$ data, we use a technique inspired by Generative Query Networks (Eslami et al., 2018). Instead of encoding the information of a single data point $(x_n, y_n)$ into $r_n$, we provide samples from the whole class-$y_n$ manifold. That is, we compute the invariant latent as:

$$r\big(\mathcal{D}_{\text{lab}}^{y_n} \setminus \{x_n\}\big) \overset{\substack{\text{concise} \\ \text{notation}}}{=} r_{y_n} = \mathbb{E}_{x \sim \mathcal{D}_{\text{lab}}^{y_n} \setminus \{x_n\}}\big[f_{\theta_{\text{inv}}}(x)\big] \approx \frac{1}{m}\sum_{i=1}^{m} f_{\theta_{\text{inv}}}(x^i) \tag{2}$$

where $\theta_{\text{inv}}$ are the parameters of this embedding. We explicitly exclude the data point at hand $x_n$ from this expectation value; in the infinite labelled data limit, the probability of sampling $x_n$ from $\mathcal{D}_{\text{lab}}^{y_n}$ would vanish. We include the simplified notation $r_{y_n}$ for subsequent mathematical clarity.

This procedure invalidates the assumption that the data is generated i.i.d. conditioned on a set of model parameters, since $r_y$ is computed using a number of other data points $x^i$ with label $y$. For notational simplicity, we will suppress this fact, and consider the likelihood $p(x, y)$ as if it were i.i.d. per data point. It is not difficult to augment the equations that follow to incorporate the full dependencies, but we find this to obfuscate the discussion (see Appendix B for full derivations). We ignore the bias introduced from the non-i.i.d. generation process; this could be avoided by holding out a dedicated labelled data set (Garnelo et al., 2018a;b), but we find it empirically insignificant.

The primary purpose of our approach to the invariant representation used in Equation 2 is to provide exactly the information needed to learn a global-class embedding: namely, to learn what the elements of the class manifold have in common. However, our approach provides a secondary advantage. During training we will use values of $m$ (see Equation 2) sampled uniformly between 1 and some small maximal value $m_{\text{max}}$. The $r_y$ embedding will thus learn to work well for various values of $m$, including $m = 1$. Consequently, at inference time, any unlabelled data point $x$ can be immediately embedded via $f_{\theta_{\text{inv}}}(x)$. This is ideal for downstream usage; we will use this technique in Section 3.1 to competitively classify unlabelled test-set data using only $f_{\theta_{\text{inv}}}(x)$.

In order to approximate the integral in Equation 1 over the equivariant latent, we use variational inference following the standard VAE approach (Kingma & Welling, 2014; Rezende et al., 2014):

$$q_{\phi_{\text{cov}}}(v|r_y, x) = \mathcal{N}\big(\mu_{\phi_{\text{cov}}}(r_y, x), \sigma_{\phi_{\text{cov}}}^2(r_y, x)I\big) \tag{3}$$

where $\phi_{\text{cov}}$ are the parameters of the variational distribution over the equivariant latent. Note that $v$ is inferred from $r_y$ (and $x$), not $y$, since $r_y$ is posited to be a multi-dimensional latent vector that represents the rich set of global properties of the class $y$, rather than just its label. As is shown empirically in Section 3, $v$ only learns to store the intra-class variations, rather than the class-label information. This is because the class-label information is easier to access directly from the deterministic representation $r_y$, rather than indirectly through the stochastic $v$. We choose to provide both $r_y$ and $x$ to $q_{\phi_{\text{cov}}}(v|r_y, x)$ in order to provide the variational distribution with more flexibility.

We thus arrive at a lower bound on $\log p(x, y)$ given in Equation 1 following the standard arguments:

$$\mathcal{L}_{\text{lab}} = \mathbb{E}_{q(v|r_y, x)} \log p(x|r_y, v) - D_{\text{KL}}\big[q(v|r_y, x)\big|\big|p(v)\big] + \log p(y) \tag{4}$$

where the various model parameters are suppressed for clarity.

The intuition associated with the inference of $r_y$ from multiple same-class, but complementary data points, as well as the inference of $v$ from $r_y$ and $x$ is depicted heuristically in Figure 1.

EQUIVAE is designed to learn a representation that stores global-class information, and as such, it can be used for semi-supervised learning. Thus, an objective function for unlabelled data must be specified to accompany the labelled-data objective function given in Equation 4.

We marginalise over the label in $p(x, y)$ (Equation 1) when a data point is unlabelled. In order to perform variational inference in this case, we use a variational distribution of the form:

$$q(v, y|x) = q_{\phi_{\text{cov}}}(v|r_y, x)\, q_{\phi_{y\text{-post}}}(y|x) \tag{5}$$

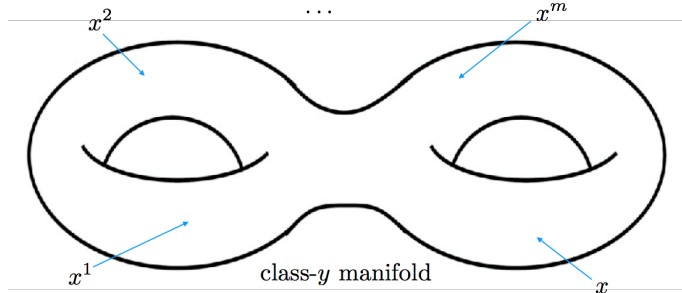

$\{x^1, x^2, \ldots, x^m\} \longrightarrow r_y$
(embedding global class-manifold information)

$x, r_y \longrightarrow v$
(embedding info. associated with instance $x$ given class)

Figure 1: Heuristic depiction of class-$y$ manifold. The invariant latent $r_y$ will encode global-manifold information, whereas equivariant latent $v$ will encode coordinates of $x$ on the manifold.

where $q_{\phi_{\mathrm{cov}}}(v|r_y, x)$ is the same distribution as is used in the labelled case, given in Equation 3. The unlabelled setting requires an additional inference distribution to infer the label $y$, which is achieved with $q_{\phi_{y\text{-post}}}(y|x)$, parametrised by $\phi_{y\text{-post}}$. Once $y$ is inferred from $q_{\phi_{y\text{-post}}}(y|x)$, $r_y$ can be deterministically calculated using Equation 2 from the labelled data set $\mathcal{D}_{\mathrm{lab}}^y$ for class $y$, of which $x$ is no longer a part. With $r_y$ and $x$, the equivariant latent $v$ is inferred via $q_{\phi_{\mathrm{cov}}}(v|r_y, x)$.

Using this variational inference procedure, we arrive at a lower bound for $\log p(x)$:

$$\mathcal{L}_{\mathrm{unlab}} = \mathbb{E}_{q(y|x)}\Big[\mathbb{E}_{q(v|r_y, x)}\log p(x|r_y, v) - D_{\mathrm{KL}}\big[q(v|r_y, x)\big|\big|p(v)\big]\Big] - D_{\mathrm{KL}}\big[q(y|x)\big|\big|p(y)\big] \quad (6)$$

where the model parameters are again suppressed for clarity.

We will compute the expectation over the discrete distribution $q(y|x)$ in Equation 6 exactly in order to avoid the problem of back propagating through discrete variables. However, this expectation could be calculated by sampling using standard techniques (Brooks et al., 2011; Jang et al., 2017; Maddison et al., 2017).

Therefore, the evidence lower bound objective for semi-supervised learning becomes:

$$\sum_{x,\ \mathrm{unlab.}} \log p(x) + \sum_{(x,y),\ \mathrm{lab.}} \log p(x, y) \ \geq \ \mathcal{L}_{\mathrm{semi}} \ = \ \sum_{x,\ \mathrm{unlab.}} \mathcal{L}_{\mathrm{unlab}} + \sum_{(x,y),\ \mathrm{lab.}} \mathcal{L}_{\mathrm{lab}} \quad (7)$$

In order to ensure that $q_{\phi_{y\text{-post}}}(y|x)$ does not collapse into the local minimum of predicting a single label for every $x$ value, we add $\log q_{\phi_{y\text{-post}}}(y|x)$ to $\mathcal{L}_{\mathrm{lab}}$. This is done in Kingma et al. (2014) and Siddharth et al. (2017), however, we do not add any hyperparameter in front of this term unlike those works. We also do not add a hyperparameter up-weighting $\mathcal{L}_{\mathrm{lab}}$ overall, as is done in Siddharth et al. (2017). The only hyperparameter tuning we perform is to choose latent dimensionality (either 8 or 16) and to choose $m_{\mathrm{max}}$, between 1 and which $m$ (see Equation 2) varies uniformly during training.

## 3 RESULTS

We carry out experiments on both the MNIST data set (LeCun et al., 1998) and the Street View House Numbers (SVHN) data set (Netzer et al., 2011). These data sets are appropriately modelled with EQUIVAE, since digits from a particular class live on a smooth, a-priori-unknown manifold.

EQUIVAE requires some labelled data. Forcing the model to reconstruct $x$ through a representation $r_y$ that only has access to other members of the $y$ class is what forces $r_y$ to represent the common information of that class, rather than a representation of the particular instantiation $x$. Thus, the requirement of some labelled data is at the heart of EQUIVAE. Indeed, we trained several versions of the EQUIVAE generative model in the unsupervised setting, allowing $r$ to receive $x$ directly. The results were as expected: the equivariant latent is completely unused, with the model unable to reconstruct the structure in each class, as it essentially becomes a deterministic autoencoder.

In Section 3.1, we study the invariant-equivariant properties of the representations learnt with EQUIVAE in the supervised setting, where we have access to the full set of labelled training data. The semi-supervised learning setting is discussed in Section 3.2. The details of the experimental setup used in this section are provided in Appendix A.

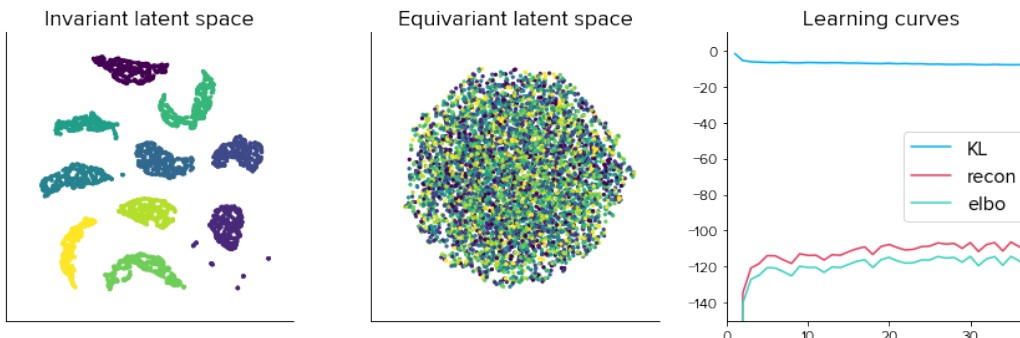

Figure 2: Validation-set results for EQUIVAE on MNIST. Invariant (left) and equivariant (middle) latent representations are shown reduced to 2D using UMAP. Learning curves are shown (right) with the ELBO broken down into the sum of the reconstruction and (negative) KL terms, as in Equation 4.

## 3.1 SUPERVISED LEARNING

With the full training data set labelled, EQUIVAE is able to learn to optimise both equivariant and invariant representations at every training step. The supervised EQUIVAE (objective given in Equation 4) converges in approximately 40 epochs on the MNIST training datset of 55,000 data points and in 90 epochs on the SVHN training datset of 70,000 data points.

The training curves on MNIST are shown on the right in Figure 2. The equivariant latent is learning to represent non-trivial information from the data as evidenced by the KL between the equivariant variational distribution and its prior not vanishing at convergence.

However, when visualised in 2 dimensions using UMAP for dimensional reduction (McInnes & Healy, 2018), the equivariant latent $v$ appears not to distinguish between digit classes, as is seen in the uniformity of the class-coloured labels in the middle plot of Figure 2. The apparent uniformity of $v$ reflects two facts: the first is that, given that the generative model gets access to another latent containing the global class information, the equivariant latent does not need to distinguish between classes. The second is that the equivariant manifolds should be similar across all MNIST digits. Indeed, they all include rotations, stretches, stroke thickness, among smooth transformations.

Finally, on the left in Figure 2, the invariant representation vectors $r_y$ are shown, dimensionally reduced to 2 dimensions using UMAP, and coloured according to the label of the class. These representations are well separated for each class. Each class has some spread in its invariant representations due to the fact that we choose relatively small numbers of complementary samples, $m$ (see Equation 2). The model shown in Figure 2 had $m$ randomly selected between 1 and 7 during training, with $m = 5$ used for visualisation. The outlier points in this plot are exaggerated by the dimensional reduction; we show this below in Figure 4 by considering a EQUIVAE with 2D latent.

The SVHN results are similar to Figure 2, with slightly less uniformity in the equivariant latent.

All visualisations in this work, including those in Figure 2, use data from the validation set (5,000 images for MNIST; 3,257 for SVHN). We reserve the test set (10,000 images for MNIST; 26,032 for SVHN) for computing the accuracy values provided. We did not look at this test set during training and hyperparameter tuning. In terms of hyperparameter tuning, we only tuned the number of epochs for training, the range of $m$ values (i.e. $m_{\max} = 7$ for MNIST; $m_{\max} = 10$ for SVHN), and chose between 8 and 16 for the dimensionality of both latents (16 chosen for both data sets). We fixed the architecture at the outset to have ample capacity for this task, but did not vary it in our experiments.

In order to really see what information is stored in the equivariant and invariant latents, we consider them in the context of the generative model. We show reconstructed images in various latent-variable configurations in Figure 3. To show samples from the equivariant prior $p(v)$ we fix a single invariant representation $r_y$ for each class $y$ by taking the mean $r_y$ over each class in the validation set. On the left in Figure 3 we show random samples from $p(v)$ reconstructed along with $r_y$ for each class $y$ ascending from 0 to 9 in each column. Two properties stand out from these samples: firstly, the

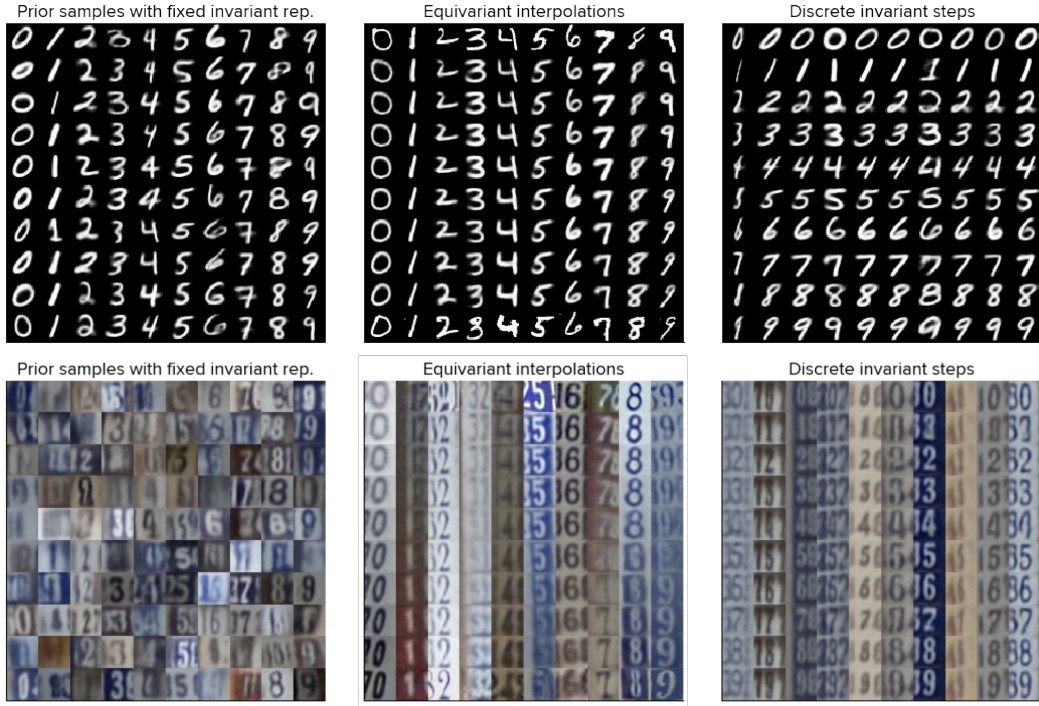

Figure 3: Generated images (left) sampled from the prior $p(v)$ for each $r_y$, (middle) reconstructed from equivariant interpolations between the embeddings of same-class digits with fixed $r_y$, and (right) reconstructed from latent pairs $(r_y^i, v^j)$ where $(r_y^i, v^i)$ is an encoded image with $y = i$.

samples are (almost) all from the correct class for MNIST (SVHN), showing that the invariant latent is representing the class information well. Secondly, there is appreciable variance within each class, showing that samples from the prior $p(v)$ are able to represent the intra-class variations.

The middle plot in Figure 3 shows interpolations between actual digits of the same class (the top and bottom rows of each subfigure), with the invariant representation fixed throughout. These interpolations are smooth, as is expected from interpolations of a VAE latent, and cover the trajectory between the two images well. This again supports the argument that the equivariant representation $v$, as a stochastic latent variable, is appropriate for representing the smooth intra-class transformations.

To create the right-hand side of Figure 3, a validation-set image for each digit $i$ is encoded to create the set of latents $\{r_{y=i}^i, v^i\}_{i=0}^9$, from which we reconstruct images using the latent pairs $(r_{y=i}^i, v^j)$ for $i, j = 0, \ldots, 9$. Thus we see in each row a single digit and in each column a single style. It is most apparent in the SVHN results that the equivariant latent controls all stylistic aspects of the image, including the non-central digits, whereas the invariant latent controls only the central digit.

In Figure 4 we show an EQUIVAE trained with 2-dimensional invariant and equivariant latents on MNIST for clearer visualisation of the latent space. On the right, reconstructions are shown with fixed $r_y$ for each $y = 3, 4, 5, 6$ and with the identical set of evenly spaced $v$ over a grid spanning from $-2$ to $+2$ in each coordinate (2 prior standard deviations). The stylistic variations appear to be similar for the same values of $v$ across different digits, as was partially evidence on the right of Figure 3. On the left of Figure 4 we see where images in the validation set are encoded, showing significant distance between clusters when dimensional reduction is not used. Finally, in the middle of Figure 4 we see the evenly-spaced reconstruction of the full invariant latent space. This shows that, though the invariant latent is not storing stylistic information, it does contain the relative similarity of the base version of each digit. This lends justification to our assertion that the invariant latent $r_y$ represents more information that just the label $y$.

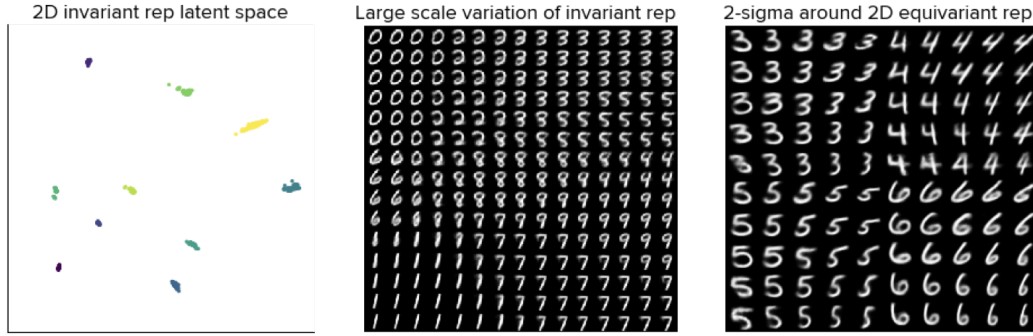

Figure 4: Latent variables for EQUIVAE with 2 dimensional latents. The invariant latent space (left), reconstructions from evenly-spaced variations of $r_y$ covering the full space at fixed $v = \vec{0}$ (middle), and reconstructions from 2-prior-standard-deviation variations of $v$ at fixed $r_y$ (right) are shown.

Table 1: Supervised error rates on MNIST (10,000 images) and SVHN (26,032 images) test sets.

|  | Technique | Error rate |
|---|---|---|
| **MNIST** | EQUIVAE Benchmark neural classifier | $0.84 \pm 0.03$ |
|  | EQUIVAE (neural classifier using $f_{\theta_{\mathrm{inv}}}(x)$) | $0.82 \pm 0.03$ |
|  | EQUIVAE (distance based on $f_{\theta_{\mathrm{inv}}}(x)$) | $0.82 \pm 0.05$ |
|  | Stacked VAE (M1+M2) (Kingma et al., 2014) | $0.96$ |
|  | Adversarial Autoencoders (Makhzani et al., 2016) | $0.85 \pm 0.02$ |
| **SVHN** | EQUIVAE Benchmark neural classifier | $10.04 \pm 0.14$ |
|  | EQUIVAE (neural classifier using $f_{\theta_{\mathrm{inv}}}(x)$) | $11.97 \pm 0.34$ |
|  | EQUIVAE (distance based on $f_{\theta_{\mathrm{inv}}}(x)$) | $12.30 \pm 0.28$ |

We have thus seen that the invariant representation $r_y$ in EQUIVAE learns to represent global-class information and the equivariant representation $v$ learns to represent local, smooth, intra-class information. This is what we expected from the theoretical considerations given in Section 2.

We now show quantitatively that the invariant representation $r_y$ learns the class information by showing that it alone can predict the class $y$ as well as a dedicated classifier. In order to predict an unknown label, we employ a direct technique to compute the invariant representation $r_y$ from $x$. We simply use $f_{\theta_{\mathrm{inv}}}(x)$ from Equation 2. We can then pass $f_{\theta_{\mathrm{inv}}}(x)$ into a neural classifier, or we can find the nearest cluster mean $r_y$ from the training set and assign class probabilities according to $p(\mathrm{label}(x) = y) \propto \exp(-||f_{\theta_{\mathrm{inv}}}(x) - r_y||^2)$. We find that classifying test-set images using this 0-parameter distance metric performs as well as using a neural classifier (2-layer dense dropout network with 128, 64 neurons per layer) with $f_{\theta_{\mathrm{inv}}}(x)$ as input. Note that using $p(y|x) \propto p(x, y)$ is roughly equivalent to our distance-based classifier, as $p(y|x)$ will be maximal when $r_y$ (computed from the training data with label $y$) is most similar to $f_{\theta_{\mathrm{inv}}}(x)$. Thus, our distance-based technique is a more-direct approach to classification than using $p(y|x)$.

Our results are shown in Table 1, along with the error rate of a dedicated, end-to-end neural network classifier, identical in architecture to that of $f_{\theta_{\mathrm{inv}}}(x)$, with 2 dropout layers added. This benchmark classifier performs similarly on MNIST (slightly better on SVHN) to the simple classifier based on finding the nearest training-set cluster to $f_{\theta_{\mathrm{inv}}}(x)$. This is a strong result, as our classification algorithm based on $f_{\theta_{\mathrm{inv}}}(x)$ has no direct classification objective in its training, see Equation 4. Our uncertainty bands quoted in Table 1 use the standard error on the mean (standard deviation divided by $\sqrt{N-1}$), with $N = 5$ trials.

Results from selected, relevant works from the literature are also shown in Table 1. Kingma et al. (2014) do not provide error bars, and they only provide a fully supervised result for their most-

Table 2: Semi-supervised test-set error rates for various labelled-data-set sizes.

| | Labels | EQUIVAE | Benchmark | Siddharth et al. (2017) | Kingma et al. (2014) |
|---|---|---|---|---|---|
| **MNIST** | | | | | **(M2)** |
| | 100 | $8.90 \pm 0.70$ | $21.91 \pm 0.66$ | $9.71 \pm 0.91$ | $11.97 \pm 1.71$ |
| | 600 | $3.99 \pm 0.17$ | $6.64 \pm 0.35$ | $3.84 \pm 0.86$ | $4.94 \pm 0.13$ |
| | 1000 | $3.34 \pm 0.17$ | $5.43 \pm 0.31$ | $2.88 \pm 0.79$ | $3.60 \pm 0.56$ |
| | 3000 | $2.23 \pm 0.14$ | $2.96 \pm 0.11$ | $1.57 \pm 0.93$ | $3.92 \pm 0.63$ |
| **SVHN** | | | | | **(M1+M2)** |
| | 1000 | $37.95 \pm 0.66$ | $39.64 \pm 1.47$ | $38.91 \pm 1.06$ | $36.02 \pm 0.10$ |
| | 3000 | $24.95 \pm 0.57$ | $25.50 \pm 0.91$ | $29.07 \pm 0.83$ | — |

powerful, stacked / pre-trained VAE M1+M2, but it appears as if their learnt representation is less accurate in its classification of unlabelled data. Makhzani et al. (2016) perform slightly worse than EQUIVAE, but within error bars. Makhzani et al. (2016) train for 100 times more epochs than we do, and use over 10 times more parameters in their model, although they use shallower dense networks. Note also that Kingma et al. (2014) and Makhzani et al. (2016) train on a training set of 50,000 MNIST images, whereas we use 55,000. We are unable to compare to Siddharth et al. (2017) as they do not provide fully supervised results on MNIST, and none of these comparable approaches provide fully supervised results on SVHN.

## 3.2 SEMI-SUPERVISED LEARNING

For semi-supervised learning, we maximise $\mathcal{L}_{\text{semi}}$ given in Equation 7. Test-set classification error rates are presented in Table 2 for varying numbers of labelled data. We compare to a benchmark classifier with similar architecture to $q_{\phi_{y\text{-post}}}(y|x)$ (see Equation 5) trained only on the labelled data, as well as to similar VAE-based semi-supervised work. The number of training epochs are chosen to be 20, 25, 30, and 35, for data set sizes 100, 600, 1000, and 3000, respectively, on MNIST, and 20 and 30 epochs for data set sizes 1000, and 3000, respectively, on SVHN. We use an 8D latent space and $m_{\max} = 4$ for MNIST, and an 16D latent space and $m_{\max} = 10$ for SVHN. Otherwise, no hyperparameter tuning is performed. Each of our experiments is run 5 times to get the mean and (standard) error on the estimate of the mean in Table 2.

We find that EQUIVAE performs better than the benchmark classifier with the same architecture (plus two dropout layers appended) trained only on the labelled data, especially for small labelled data sets. Furthermore, EQUIVAE performs competitively (within error bars or better) relative to its most similar comparison, Siddharth et al. (2017), which is a VAE-based probabilistic model that treats the labels and the style of the data separately. Given its relative simplicity, rapid convergence (20-35 epochs), and lack of hyperparameter tuning performed, we consider this to be an indication that EQUIVAE is an effective approach to jointly learning invariant and equivariant representations, including in the regime of limited labelled data.

## 4 CONCLUSIONS

We have introduced a technique for jointly learning invariant and equivariant representations of data comprised of discrete classes of continuous values. The invariant representation encodes global information about the given class manifold which is ensured by the procedure of reconstructing a data point through complementary samples from the same class. The equivariant representation is a stochastic VAE latent that learns the smooth set of transformations that cover the instances of data on that class manifold. We showed that the invariant latents are so widely separated that a 99.18% accuracy can be achieved on MNIST (87.70% on SVHN) with a simple 0-parameter distance metric based on the invariant embedding. The equivariant latent learns to cover the manifold for each class of data with qualitatively excellent samples and interpolations for each class. Finally, we showed that semi-supervised learning based on such latent variable models is competitive with similar approaches in the literature with essentially no hyperparameter tuning.

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

## A  EXPERIMENTAL SETUP

In this appendix we provide details of the experimental setup that was used to generate the results from Section 3.

For our implementation of EQUIVAE, we use relatively standard neural networks. All of our experiments use implementations with well under 1 million parameters in total, converge within a few hours (on a Tesla K80 GPU), and are exposed to minimal hyperparameter tuning.

In particular, for the deterministic class-representation vector $r_y$ given in Equation 2, we parametrise $f_{\theta_{\text{inv}}}(x)$ using a 5-layer, stride-2 (stride-1 first layer), with 5x5 kernal size, convolution network, followed by a dense hidden layer. The mean of these $m$ embeddings $f_{\theta_{\text{inv}}}(x_y^i)$ is taken, followed then by another dense hidden layer, and the final linear dense output layer. This is shown for a $y = 6$ MNIST digit in the top shaded box of Figure 5. Our implementation uses $(8, 16, 32, 64, 64)$ filters in the convolution layers, and $(128, 64)$ hidden units in the two subsequent dense layers for a 16 dimensional latent (the number of units in the dense layers are halved when using 8 dimensional latents, as in our semi-supervised experiments on MNIST).

We parametrise the approximate posterior distribution $q_{\phi_{\text{cov}}}(v|r_y, x)$ over the equivariant latent as a diagonal-covariance normal distribution, $\mathcal{N}(\mu_{\phi_{\text{cov}}}(r_y, x), \sigma^2_{\phi_{\text{cov}}}(r_y, x))$, following the SGVB algorithm (Kingma & Welling, 2014; Rezende et al., 2014). For $\mu_{\phi_{\text{cov}}}(r_y, x)$ and $\sigma^2_{\phi_{\text{cov}}}(r_y, x)$, we use the identical convolution architecture as for the invariant embedding network as an initial embedding for the data point $x$. This embedding is then concatenated with the output of a single dense layer

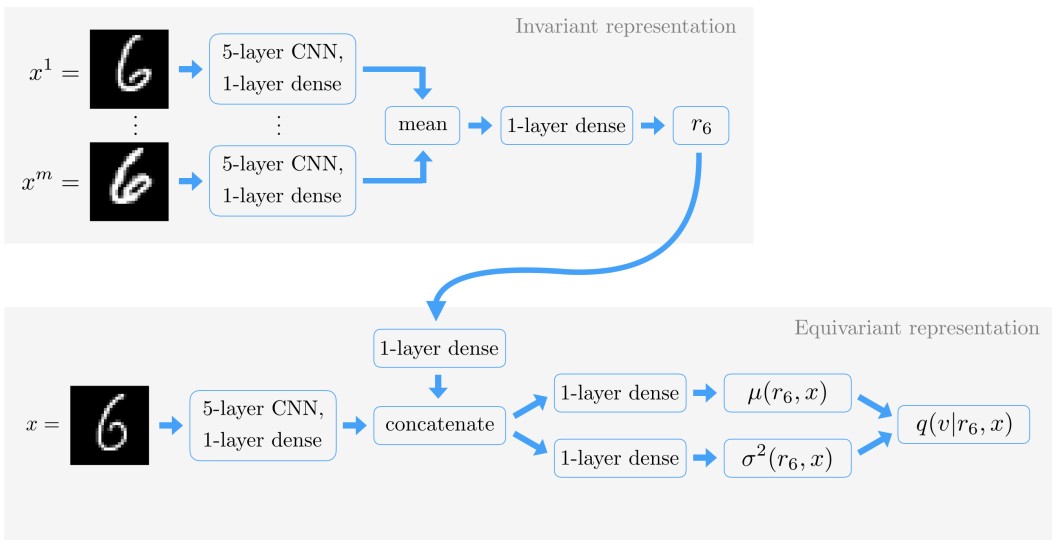

Figure 5: Example of variational encoding of a particular MNIST digit $x$ (from the 6 class) both in terms of its invariant representation (top) and its equivariant representation (bottom).

that transforms $r_y$, the output of which is then passed to one more dense hidden layer for each $\mu$ and $\sigma^2$ separately. This is shown in the bottom shaded box of Figure 5.

The generative model $p_\theta(x|r_y, v)$ is based on the DCGAN-style transposed convolutions (Radford et al., 2016), and is assumed to be a Bernoulli distribution for MNIST (Gaussian distribution for SVHN) over the conditionally independent image pixels. Both the invariant representation $r_y$ and the equivariant representation $v$, are separately passed through a single-layer dense network before being concatenated and passed through another dense layer. This flat embedding that combines both representations is then transpose convolved to get the output image in a way the mirrors the 5-layer convolution network used to embed the representations in the first place. That is, we use $(64, 128)$ hidden units in the first two dense layers, and then $(64, 32, 16, 8, n_{\text{colours}})$ filters in each transpose convolution layer, all with 5x5 kernals and stride 2, except the last layer, which is a stride-1 convolution layer (with padding to accommodate different image sizes).

In our semi-supervised experiments, we implement $q_{\phi_{y\text{-post}}}(y|x)$ using the same (5-CNN, 1-dense) encoding block to provide an initial embedding for $x$. This is then concatenated with stop_grad$(f_{\theta_{\text{inv}}}(x))$ and passed to a 2-layer dense dropout network with $(128, 64)$ units. The use of stop_grad$(f_{\theta_{\text{inv}}}(x))$ is simply that $f_{\theta_{\text{inv}}}(x)$ is learning a highly relevant, invariant representation of $x$ that $q_{\phi_{y\text{-post}}}(y|x)$ might as well get access to. However, we do not allow gradients to pass through this operation since $f_{\theta_{\text{inv}}}(x)$ is meant to learn from the complementary data of known same-class members only.

As discussed in Section 2, the number of complementary samples $m$ used to reconstruct $r_y$ (see Equation 2) is chosen randomly at each training step in order to ensure that $r_y$ is insensitive to $m$. For our supervised experiments where labelled data are plentiful, $m$ is randomly select between 1 and $m_{\text{max}}$ with $m_{\text{max}} = 7$ for MNIST ($m_{\text{max}} = 10$ for SVHN), whereas in the semi-supervised case $m_{\text{max}} = 4$ for MNIST ($m_{\text{max}} = 10$ for SVHN).

We perform standard, mild preprocessing on our data sets. MNIST is normalised so that each pixel value lies between 0 and 1. SVHN is normalised so that each pixel has zero mean and unit standard deviation over the entire dataset.

Finally, all activation functions that are not fixed by model outputs are taken to be rectified linear units. We use Adam (Kingma & Ba, 2015) for training with default settings, and choose a batch size of 32 at the beginning of training, which we double successively throughout training.

## B   DERIVATION OF LIKELIHOOD LOWER BOUNDS

In this appendix we detail the derivations of the log-likelihood lower bounds that were provided in Section 2.

EQUIVAE is relevant when a non-empty set of labelled data is available. We write the data set as

$$\mathcal{D} = \mathcal{D}_{\text{lab}} \cup \mathcal{D}_{\text{unlab}} = \{x_n, y_n\}_{n=1}^{N_{\text{lab}}} \cup \{x_n\}_{n=1}^{N_{\text{unlab}}} \tag{8}$$

We also decompose $\mathcal{D}_{\text{lab}} = \cup_y \mathcal{D}_{\text{lab}}^y$, where $\mathcal{D}_{\text{lab}}^y$ is the set of labelled instantiations $x$ with label $y$. In particular, in what follows we think of $\mathcal{D}_{\text{lab}}^y$ as containing only the images $x$, not the labels, since they are specified by the index on the set. We require at least two labelled data points from each class, so that $|\mathcal{D}_{\text{lab}}^y| \geq 2 \; \forall y$.

We would like to maximise the log likelihood that our model generates both the labelled and the unlabelled data, which we write as:

$$\log p(\mathcal{D}) = \log p(\mathcal{D}_{\text{lab}}) + \log p(\mathcal{D}_{\text{unlab}}|\mathcal{D}_{\text{lab}}) \tag{9}$$

where we make explicit here the usage of labelled data in the unlabelled generative model.

For convenience, we begin by repeating the generative model for the labelled data in Equation 1, except with the deterministic integral over $r_n$ completed:

$$p\big(\{x_n, y_n\}_{n=1}^{N_{\text{lab}}}\big) = \prod_{n=1}^{N_{\text{lab}}} \int dv_n \, p_\theta\big(x_n | r(\mathcal{D}_{\text{lab}}^{y_n} \setminus \{x_n\}), v_n\big) \, p(v_n) \, p(y_n) \tag{10}$$

We will simplify the notation by writing $\widehat{x_n} = \mathcal{D}_{\text{lab}}^{y_n} \setminus \{x_n\}$ and $r_{y_n, \widehat{x_n}} = r(\mathcal{D}_{\text{lab}}^{y_n} \setminus \{x_n\})$, but keep all other details explicit.

We seek to construct a lower bound on $\log p(\{x_n, y_n\}_{n=1}^{N_{\text{lab}}})$, namely the log likelihood of the labelled data, using the following variational distribution over $v_n$ (Equation 3):

$$q_{\phi_{\text{cov}}}(v_n | r_{y_n, \widehat{x_n}}, x_n) = \mathcal{N}\big(\mu_{\phi_{\text{cov}}}(r_{y_n, \widehat{x_n}}, x_n), \sigma^2_{\phi_{\text{cov}}}(r_{y_n, \widehat{x_n}}, x_n)I\big) \tag{11}$$

Indeed,

$$\log p\big(\{x_n, y_n\}_{n=1}^{N_{\text{lab}}}\big) = \sum_{n=1}^{N_{\text{lab}}} \log \mathbb{E}_{q_{\phi_{\text{cov}}}(v_n | r_{y_n, \widehat{x_n}}, x_n)} \frac{p_\theta\big(x_n | r_{y_n, \widehat{x_n}}, v_n\big) \, p(v_n) \, p(y_n)}{q_{\phi_{\text{cov}}}(v_n | r_{y_n, \widehat{x_n}}, x_n)}$$

$$\overset{\text{Jensen's}}{\geq} \sum_{n=1}^{N_{\text{lab}}} \mathbb{E}_{q_{\phi_{\text{cov}}}(v_n | r_{y_n, \widehat{x_n}}, x_n)} \log \frac{p_\theta\big(x_n | r_{y_n, \widehat{x_n}}, v_n\big) \, p(v_n) \, p(y_n)}{q_{\phi_{\text{cov}}}(v_n | r_{y_n, \widehat{x_n}}, x_n)} \tag{12}$$

$$= \sum_{n=1}^{N_{\text{lab}}} \mathbb{E}_{q_{\phi_{\text{cov}}}(v_n | r_{y_n, \widehat{x_n}}, x_n)} \log p_\theta\big(x_n | r_{y_n, \widehat{x_n}}, v_n\big) - D_{\text{KL}}\big[q_{\phi_{\text{cov}}}(v_n | r_{y_n, \widehat{x_n}}, x_n) \big| \big| p(v_n)\big] + \log p(y_n)$$

Which coincides with the notationally simplified lower bound objective function given in Equation 4.

We now turn to the lower bound on the unlabelled data. To start, we marginalise over the labels on the unlabelled dataset:

$$p\big(\{x_n\}_{n=1}^{N_{\text{unlab}}} \big| \mathcal{D}_{\text{lab}}\big) = \prod_{n=1}^{N_{\text{unlab}}} \sum_{y_n} \int dv_n \, p_\theta\big(x_n | r(\mathcal{D}_{\text{lab}}^{y_n}), v_n\big) \, p(v_n) \, p(y_n) \tag{13}$$

where we no longer need to remove $x_n$ from $\mathcal{D}_{\text{lab}}^{y_n}$ in $r(\cdot)$ since for the unlabelled data, $x_n \notin \mathcal{D}_{\text{lab}}^{y_n}$.

As was done for the labelled data, we construct a lower bound using variational inference. However, in this case, we require a variational distribution over both $y_n$ and $v_n$. We take:

$$q(v_n, y_n | x_n, \mathcal{D}_{\text{lab}}) = q_{\phi_{\text{cov}}}\big(v_n | r(\mathcal{D}_{\text{lab}}^{y_n}), x_n\big) \, q_{\phi_{y\text{-post}}}(y_n | x_n) \tag{14}$$

which gives

$$
\begin{aligned}
\log p\big(\{x_n\}_{n=1}^{N_{\text{unlab}}}\big|\mathcal{D}_{\text{lab}}\big) = \log \prod_{n=1}^{N_{\text{unlab}}} &\mathbb{E}_{q_{\phi_{y\text{-post}}}(y_n|x_n)}\mathbb{E}_{q_{\phi_{\text{cov}}}(v_n|r(\mathcal{D}_{\text{lab}}^{y_n}),x_n)}\Bigg[ \\
&\frac{p_\theta\big(x_n|r(\mathcal{D}_{\text{lab}}^{y_n}),v_n\big)\,p(v_n)\,p(y_n)}{q_{\phi_{\text{cov}}}\big(v_n\big|r(\mathcal{D}_{\text{lab}}^{y_n}),x_n\big)\,q_{\phi_{y\text{-post}}}(y_n|x_n)}\Bigg] \\
\overset{\text{Jensen's}}{\geq} \sum_{n=1}^{N_{\text{unlab}}} &\mathbb{E}_{q_{\phi_{y\text{-post}}}(y_n|x_n)}\mathbb{E}_{q_{\phi_{\text{cov}}}(v_n|r(\mathcal{D}_{\text{lab}}^{y_n}),x_n)}\log\frac{p_\theta\big(x_n|r(\mathcal{D}_{\text{lab}}^{y_n}),v_n\big)\,p(v_n)\,p(y_n)}{q_{\phi_{\text{cov}}}\big(v_n\big|r(\mathcal{D}_{\text{lab}}^{y_n}),x_n\big)\,q_{\phi_{y\text{-post}}}(y_n|x_n)} \\
= \sum_{n=1}^{N_{\text{unlab}}} &\mathbb{E}_{q_{\phi_{y\text{-post}}}(y_n|x_n)}\Bigg[\mathbb{E}_{q_{\phi_{\text{cov}}}(v_n|r(\mathcal{D}_{\text{lab}}^{y_n}),x_n)}\log p_\theta\big(x_n|r(\mathcal{D}_{\text{lab}}^{y_n}),v_n\big) \\
&- D_{\text{KL}}\big[q_{\phi_{\text{cov}}}(v_n|r(\mathcal{D}_{\text{lab}}^{y_n}),x_n)\big|\big|p(v_n)\big]\Bigg] - D_{\text{KL}}\big[q_{\phi_{y\text{-post}}}(y_n|x_n)\big|\big|p(y_n)\big]
\end{aligned}
\tag{15}
$$

Thus, we have Equation 6 augmented with the notational decorations that were omitted in Section 2.

Therefore, the objective

$$
\mathcal{L} = \sum_{n=1}^{N_{\text{unlab}}} \mathcal{L}_{\text{unlab}}^{(n)} + \sum_{n=1}^{N_{\text{lab}}} \mathcal{L}_{\text{lab}}^{(n)}
\tag{16}
$$

given in Equation 7, with $\mathcal{L}_{\text{lab}}^{(n)}$ given in Equation 4 and $\mathcal{L}_{\text{unlab}}^{(n)}$ given in Equation 6, is a lower bound on the data log likelihood.

