# OpenReview forum: "Invariant-equivariant representation learning for multi-class data"
_ICLR.cc/2019/Conference_

### Official Review · AnonReviewer2 · 2018-10-21
**CoVAE**

**Rating:** 4
**Confidence:** 3

**Review:**

The paper presents a VAE that uses labels to separate the learned representation into an invariant and a covariant part. The method is validated using experiments on the MNIST dataset.

The writing in this paper is somewhat problematic. Although it is hard to put the finger on a particularly severe instance, the paper is filled with vague and hyperbolic statements. Words like "efficiently", "meaningful", "natural", etc. are sprinkled throughout to confer a positive connotation, often without having a specific meaning in their context or adding any information. Where the meaning is somewhat clear, the claims are often not supported by evidence. Sometimes the claims are so broad that it is not clear what kind of evidence could support such a claim.

A relatively large amount of space is used to explain the general concept of invariant/covariant learning, which, as a general concept, is widely understood and not novel. There are other instances of overclaiming, such as "The goal of CoVAE is to provide an approach to probabilistic modelling that enables meaningful representations [...]". In fact, CoVAE is a rather specific model(class), rather than an approach to probabilistic modelling.

The paper is at times meandering. For instance, the benefits of and motivation for the proposed approach are not simply stated in the introduction and then demonstrated in the rest of the paper, but instead the paper states some benefits and motivations, explains some technical content, mentions some more benefits, repeats some motivations stated before, etc.

Many researchers working on representation learning hope to discover the underlying learning principles that lead to representations that seem natural to a human being. In this paper, labels are used to guide the representation into the "right" representation. It is in my opinion not very surprising that one can use labels to induce certain qualities deemed desirable in the representation.

To conclude, because of the writing, limited novelty, and limited experiments, I think this paper currently does not pass the bar for ICLR.

---

> ### Author Response · Authors · 2018-11-19
> **Rewrote paper to improve writing, addressed other points as well**
>
> We thank the reviewer for their helpful feedback. The reviewer's main criticism was the writing: both the usage of vague, hyperbolic statements, as well as unnecessary discussion and meandering.
>
> On the vague, hyperbolic statements, the authors' intention was to provide clarity-by-repetition with these statements, but concede with regret in retrospect that they came across unhelpfully vague or unjustified. We have purged this language throughout the paper. There is now not a single usage of "efficient", "meaningful", or "natural" in describing our own work (we do still use such words in the Introduction when discussing the literature and representation learning in general). We have also tried to eradicate other examples of similar language (e.g. "general" in the original conclusions, EquiVAE is no longer described as a "framework"). We feel that the writing is now clearer and more objective.
>
> To improve the conciseness and linearity of the paper, we have completely removed Section 2.1, which perhaps over repeated obvious attributes of our modelling choices. Section 2.1 also contained some of the problematic language referred to above. Similarly, the semi-supervised objective function and surrounding discussion was brought from Section 3.2 into Section 2 for linearity.
>
> The reviewer also briefly mentioned limited novelty and limited experiments as weaknesses of the paper. We have addressed both of these points. Firstly, we have added 9 new references to relevant work in the literature along with discussions of how our approach is related, and indeed novel (e.g. our usage multiple same-class complementary data points for the invariant latent). Secondly, we have run all of our experiments on the Street View House Numbers (SVHN) data set in order to add robustness to our experimental results. Indeed the results on SVHN are consistent with our original results on MNIST (e.g. invariant representations are well separated, semi-supervised accuracies are competitive with Siddharth et al. (2017)) with minimal changes to our modelling setup (i.e. to accommodate different image sizes).
>
> We regret the way the paper came out originally in its writing, but with the reviewers poignant criticism on this front, we feel that the updated version of our paper is much more appropriate and clear. We hope this, as well as the additional experimental results, now justify its publication.

---

### Official Review · AnonReviewer3 · 2018-11-05
**Technically Sound, Well Written, but The Key Idea is Not Very New**

**Rating:** 5
**Confidence:** 5

**Review:**

This paper is well written, and the quality of the figures is good. In this paper, the authors propose an invariant-covariant idea, which should be dated back at least to the bilinear models. The general direction is important and should be pursued further.

However, the literature is not well addressed. Eslami et al. 2018 have been cited, but some very important and related earlier works like:
[1] Kulkarni et al. 2015, Deep Convolutional Inverse Graphics Network
[2] Cheung et al. 2015, Discovering Hidden Factors of Variation in Deep Networks
were not discussed at all. The authors should certainly make an effort to discuss the connections and new developments beyond these works. At the end of section 1, the authors argue that the covariant vector could be more general, but in fact, these earlier works can achieve further equivalence, which is much stronger than the proposed covariance.

There is also an effort to compare this work to Sabour et al. 2017 and the general capsule idea. I would like to point out, the capsule concept is a much more fine-grained what & where separation rather than a coarse-grained class & pose separation in one shot. In a hierarchical representation, what & where can appear at any level as one class can consist of several parts each with a geometrical configuration space. So the comparison of this work to the generic capsule network is only superficial if the authors can not make the proposed architecture into a hierarchical separation. Besides different capsule network papers, I found another potentially useful reference on a fine-grained separation:
[3]Goroshin et al., Learning to Linearize Under Uncertainty

In the paper, it is argued several times that the latent vector r_y contains a rich set of global properties of class y, rather than just its label and the aim is that it can learn what the elements of the class manifold have in common. But this point is not supported well since we can always make a label and this latent vector r_y equivalent by a template. I think this point could be meaningful if we look at r_y's for different y, where each of the dimension may have some semantic meaning. Additional interpretation is certainly needed.

Under equation (3), "Note that v is inferred from r_y" should be "inferred from both r_y and x", which is pretty clear from the fig 5. Related to this, I could imagine some encoder can extract the 'style' directly from x, but here both r_y and x are used. I couldn't find any guarantee that v only contains the 'style' information based on the architecture with even this additional complication, could the authors comment on this?

Equation (5) is not really a marginalization and further equation (6) may not be a lower bound anymore. This is probably a relatively minor thing and a little extra care is probably enough.

The numbers in table 2 seems a little outdated.

To conclude, I like the general direction of separating the identity and configurations. The natural signals have hierarchical structures and the class manifold concept is not general enough to describe the regularities and provide a transparent representation. Rather, it's a good starting point. If the authors could carefully address the related prior works and help us understand the unique and original contributions of this work, this paper could be considered for publication.

---

> ### Comment · AnonReviewer3 · 2018-11-05
> **Equivariance**
>
> "can achieve further equivalence" -> "can achieve further equivariance"

---

> ### Author Response · Authors · 2018-11-19
> **Significant comparison to previous work added, along with all other points addressed (1 / 2)**
>
> We appreciate deeply the reviewer's detailed feedback. The reviewer made a number of useful suggestions to improve this submission, of which we felt that the most prominent theme was improving our discussion and comparison to the literature. We discuss this first.
>
> Overall, we have included 9 new references to the literature along with discussion in Section 1, which has been significantly modified. In particular, we discuss the two important references [1] and [2] provided by the reviewer. Indeed these two references are able to separate more structure into their latent variables (for data sets that contain structured labels for different variations of the data) using a 'clamping' technique during training to force particular latent components to be disentangled (as in [1]), or by predicting the labels and using a cross-covariance term in the objective function of their deterministic autoencoder (as in [2]). Similarly, we discuss InfoGAN (Chen et al, 2016), which achieves disentanglement by adding a mutual-information maximisation term between the meaningful latent and the model generations. We consider the simplicity of our probabilistic model, trained with log-likelihood maximisation (no additional objective function hyperparameters), augmented only with a complementary batch of same-class data for each training data point, to be attractive in comparison to these other approaches.
>
> However, there are other significant differences. One major difference is that our invariant representation takes as input multiple data points that come from the same class, but are different from the reconstructed data point. Thus, our invariant representation directly learns to encode the information common to the overall class, but not the individual data point. This is why there is no need for clamping or cross-covariance / mutual information penalty terms in our objective function: there is no way for the invariant representation to store data-point specific information due to the information flow through it. In further contrast to some of the other approaches in the literature, we deliberately use a deterministic latent for the class-invariant information, and a stochastic latent for the smooth 'style' information (an idea employed by Zhu et al. (2014), for their binary neurons). This modelling choice is why we do not need to force the equivariant latent to not contain any class-level information (which is clear from our results) because it is available and easier to access from the deterministic latent. This last point was a question raised by the reviewer which we have addressed with an added discussion after Equation 3 in Section 2.
>
> ...

---

> > ### Author Response · Authors · 2018-11-19
> > **Significant comparison to previous work added, along with all other points addressed (2 / 2)**
> >
> > ...
> >
> > We consider the usage of multiple data points (from the same class) in order to reconstruct a single data point to be novel as compared to previous work on the problem of learning disentangled or structured latents. To that end, we also included references to the recent Neural Processes work (Garnelo et al., 2018a;b) which also models a data set using multiple data points at once, though their goal of better modelling uncertainty in neural networks is very different from ours.
> >
> > The reviewer also pointed out that the hierarchical nature of the capsule networks approach makes our comparison to Sabour et al. (2017) superficial. We concede this point, and have removed the direct comparisons to capsule networks from the paper in order to avoid misleading the reader. This is considered justified as, though this work is in the same vein at an abstract level, the capsule networks work is sufficiently unrelated to our work.
> >
> > In order to improve the support of our claim that the invariant latent r_y contains more information than just the label y, we have updated the presentation of our results. Figure 3 now shows on the right the effect of varying r_y over each y for fixed equivariant latent v. Most importantly, Figure 4 (middle) shows the large-scale variations of r_y over the entire 2D space (recall these are results for 2D r_y). From this figure it can be seen that there is little intra-digit variation, but r_y is storing the full global relationships between the 10 MNIST digits, with 8 in the middle and the most similar digits to 8 nearby (e.g. 4 occupies a small region of r_y space immediately adjacent to 9 and 5 on either sides). We have updated the discussion around this figure to improve the justification for the claim that r_y stores more information than just its label value y.
> >
> > On the final 3 specific points made by the reviewer, we have: 1) updated the text under equation (3) to address the point that v is inferred from both r_y and x, 2) we have included an additional appendix to derive the lower bound associated with equation (6), and 3) we have added a short discussion of why the numbers in table 2 were used.
> >
> > In conclusion, we greatly appreciate the feedback, and have attempted to carefully address all the points provided. We have significantly elaborated the discussion of the literature, clarified the aspects of our approach that are novel as well as  those that are not, corrected / clarified each small suggestion made by the reviewer, and we have included results on a new, more challenging data set (Street View House Numbers). We hope these changes are sufficient to justify publication of this work in ICLR.

---

### Official Review · AnonReviewer1 · 2018-11-06
**Nice, clean generative model, wonder how it would perform on a more challenging dataset.**

**Rating:** 7
**Confidence:** 2

**Review:**

The paper proposes a genarative model for images which explicitly separates the within class variation
(the covariant part) from the across class variation (invariant part). Functionally, this achieves a similar
result as various recent works on incorporating invariances in neural nets, but the fact that it is able
to explicitly construct models for both parts of the distribution is nice. Results on MNIST are good,
but of course this is a very simple dataset. It would be very interesting to see how the model
performs on a more realistic problem.

Admittedly, I am not an expert in generative models. This is a clean paper with a clear goal, it is hard
for me to judge how original the idea is.

"Covariant" might not be the best word to use here because it has a very specific meaning in the context
of some other neural networks related to how quantities transform according to representations of a symmetry
group. This is a potential source of confusion.

---

> ### Author Response · Authors · 2018-11-19
> **Added experiments on new data set (SVHN)**
>
> We appreciate the reviewer's feedback, and have addressed both points made.
>
> Most importantly, we have included new results for all of the original experiments on the Street View House Numbers data set, which is a much more challenging data set than MNIST. These results are consistent with the MNIST results originally presented, but add robustness to the conclusions of the paper.
>
> Secondly, the word "covariant" has been replaced throughout the paper with "equivariant" in order to avoid unnecessary confusion.

---

### Meta-Review · Area_Chair1 · 2018-12-19
**A new ad-hoc approach for learning class-equivariant representations, with limited justification, literature context, and experiments.**

**Confidence:** 4
**Recommendation:** Reject

**Metareview:**

The paper presents a new approach to learn separate class-invariant and class-equivariant latent representations, by training on labeled (and optional additional unlabelled) multi class data. Empirical results on MNIST and SVHN show that the method works well.
Reviewers initially highlighted the following weaknesses of the paper: insufficient references and contrasting with related work (given that this problem space has been much explored before),
limited novelty of the approach, limited experiments (MNIST only). One reviewer also mentioned a sometimes vague, overly hyperbolic, and meandering writeup.

Authors did a commendable effort to improve the paper based on the reviews, adding new references, removing and rewriting parts of the paper to make it more focused, and providing experimental results on an additional dataset (SVHN). The paper did improve as a result. But while attenuated, the initial criticisms remain valid: the literature review and discussion remains short and too superficial. The peculiarities of the approach which grant it (modest) originality are insufficiently (theoretically and empirically) justified and not clearly enough put in context of the whole body of prior work. Consequently the proposed approach feels very ad-hoc. Finally the additional experiments are a step in the right direction, but experiments on only MNIST and SVHN are hardly enough in 2018 to convince the reader that a method has a universal potential and is more generally useful. Given the limited novelty, and in the absence of theoretical justification, experiments should be much more extensive, both in diversity of data/problems, and in the range of alternative approaches compared to, to build a convincing case.